# Diff-StyGS: 3D Gaussian Splatting Stylization via Tuning-Free Multi-View Sparse Diffusion

## Abstract

Realistic stylization in 3D Gaussian Splatting (3DGS) faces critical challenges due to restricted cross-modal style inputs (text/image) and the difficulty of preserving multi-view consistency without sacrificing efficiency. Existing methods either depend on fine-tuned conditional diffusion models (e.g., InstructPix2Pix) or require style-specific losses and latents. In this paper, we propose Diff-StyGS, a novel framework enabling 3D style transfer with multimodal inputs for pre-trained 3DGS via tuning-free Stable Diffusion (SD). Our approach introduces multi-view stylized attention by dual attention control in SD with (i) Style-Infused Attention (SIA) and (ii) Multi-View Adaptive Sparse Attention via Shared-Query (MASA-SQ). Specifically, SIA decouples content by reusing 3DGS-rendered query features while adjusting style based on stylized keys and values from SD. MASA-SQ reduces cross-view inconsistency and computational overheads through adaptive fusion of style and sparsity-aware multi-view priors. Furthermore, we present the Wavelet Frequency Alignment Loss for stylized distribution alignments across frequency domains. To further accelerate style optimization, we leverage a 3D sparse-view strategy to select geometrically representative views through Maximin Distance Design. Extensive experiments demonstrate that Diff-StyGS outperforms state-of-the-art text/image-based 3DGS style transfer methods in terms of multi-view consistency, stylization quality, and content fidelity.

## 1 Introduction

Recent advancements in 2D generation have demonstrated convincing success in content creation with Diffusion Models (DM) (Wang et al., 2025; Wei et al., 2023; Yin et al., 2024; Li et al., 2025; Cheng et al., 2025). In addition, high-quality 3D reconstruction is becoming more accessible by radiance field-related approaches (Barron et al., 2021; Deng et al., 2022; Lee et al., 2024; Yao et al., 2025). Although these DMs are superior in image editing (Avrahami et al., 2022; Brooks et al., 2023; Chen et al., 2023a; Kawar et al., 2023; Pan et al., 2023) through exceptional 2D generative priors and flexible conditions, their 3D awareness is fundamentally lacking, leading to multi-view inconsistencies when naively applied to 3D content (Lin et al., 2025; Yang et al., 2025). Conversely, modern 3D representations, such as NeRF (Barron et al., 2022; 2023; Mildenhall et al., 2020) and 3D Gaussian Splatting (3DGS) (Kerbl et al., 2023; Keyang et al., 2024; Kwak et al., 2025), achieve realistic novel view synthesis through differentiable rendering. However, direct editing of these compressed neural latents and 3D assets remains a non-trivial task.

In particular, 3D style transfer faces the following challenges: (i) (**Expansive Controllability**) Common 3D style transfer methods only support one single input modality, either text or a reference image for stylization. Furthermore, although recent conditioning mechanisms, such as Control-Net (Zhang et al., 2023), adapter modules (Mou et al., 2023; Ye et al., 2023; Zhuang et al., 2024), and InstructPix2Pix (Brooks et al., 2023) improve conditional controllability, they often require extensive fine-tuning with substantial computational and memory resources. (ii) (**Limited Fidelity**) One typical strategy attempts direct 3D space manipulations (Yuan et al., 2022; Kovács et al., 2024; Mei et al., 2024; Liu et al., 2024c; 2025) via style-specific training objectives and embedded features from style images. However, it struggles with arbitrary styles and is prone to producing visual artifacts (StyleGaussian (Liu et al., 2024c) as shown in Fig. 4), leading to limited generative fidelity. (iii) (**Multi-view Inconsistency**) Another prevalent paradigm is iterative tuning by alternations between 2D DM-based editing (Chen et al., 2024c; Haque et al., 2023) of rendered views and 3D reconstruc-

tion updates (Zhang et al., 2024; 2022; Kong et al., 2025). However, it inherits DMs' stochastic nature across views, which generates inconsistent appearances between multiple views. (iv) (**Massive Computations**) The above methods typically require extensive, complicated training to implement 3D stylization, such as iterative alternating tuning between 2D editing and 3D reconstruction, incurring heavy computing overheads.

To address the above challenges, we propose **Diff-StyGS**, a novel pipeline for multimodal 3D style transfer that strikes a balance between editing fidelity and 3D consistency in the context of 3DGS, with superior controllability and efficiency. Our pipeline unfolds in two main stages: ① generating a set of view-aligned, consistently stylized images using attention substitution modules, and ② optimizing the 3DGS scene with content-style loss functions.

The first stage centers on consistent 2D stylization. **Style-Infused Attention (SIA)** performs training-free style injection by replacing attention *key* and *value* with style features (from DDIM inversion or text prompts) while fusing content *query* to preserve its spatial structure. To elevate this from single-view editing to an efficient 3D-coherent process, we propose **Multi-View Adaptive Sparse Attention via Shared-Query (MASA-SQ)** to enforce structural consistency by sharing the content *query* across views and adaptively fusing reference *key* and *value* with AdaIN (Huang et al., 2017). In the second stage, since conventional pixel losses (*e.g.*, $\mathcal{L}_1$, SSIM) fail to capture high-frequency textures critical for style expression in Fig. 2, we propose **Wavelet Frequency Alignment Loss (WFAL)** by decomposing rendered/stylized images into multi-scale wavelet sub-bands and enforcing distribution alignment with Sliced Wasserstein Distance (SWD). To enhance the overall efficiency, we introduce a **sparse-view selection (SVS)** strategy to select a compact yet representative set of input views.

Our method effectively facilitates 3D style transfer. Specifically, by injecting style features (from either text prompts or reference images) into DM's attention, SIA implicitly encodes content, layout, and style information in the *query*, *key*, and *value* components, leading to enhanced controllability and fidelity. MASA-SQ shares the content *query* across views, ensuring stylistically view-consistent generation. WFAL further improves fidelity by aligning frequency distributions. Meanwhile, to reduce computation costs, multiple acceleration techniques are incorporated, including sparse attention in MASA-SQ, sparse-view selection, and frozen SD in SIA and MASA-SQ without fine-tuning. Various experiments demonstrate our significant advantages in generating high-quality stylized 3D scenes across diverse scenarios compared with SOTA 3DGS stylization approaches.

Our main contributions are summarized as follows:

- We propose Diff-StyGS, a comprehensive and efficient pipeline for high-quality, multi-view consistent 3D scene stylization with both text and image style sources in a zero-shot manner.
- We introduce a novel multi-view attention module for DMs, comprising SIA for versatile style injection and MASA-SQ to ensure cross-view consistency and computational efficiency without any model training.
- We present WFAL to enable faithful style transfer to 3DGS representations by aligning multi-scale wavelet features, complemented by a Maximin Distance-based SVS for acceleration.
- Comprehensive experiments demonstrate the effectiveness of our method across various 3D scenes with multimodal style sources (text/image).

## 2 RELATED WORK

### 2.1 2D DIFFUSION-BASED GENERATION AND EDITING

2D DMs (Feng et al., 2023; Rombach et al., 2022; Saharia et al., 2022; Zhao et al., 2023) have demonstrated qualified text-guided image synthesis. Subsequent studies extend these models to various tasks, including inpainting (Corneanu et al., 2024; Liu et al., 2024b; Xie et al., 2023), style transfer (Chen et al., 2023b; Chung et al., 2024; Deng et al., 2024; Wang et al., 2023), and video generation & editing (Ceylan et al., 2023; Chen et al., 2024a; Esser et al., 2023; Gupta et al., 2024). To achieve fine-grained control, P2P (Hertz et al., 2023) replaces cross-attention layers to edit images while preserving structural coherence semantically. MasaCtrl (Cao et al., 2023) converts self-attention in DMs into the mutual one guided by masks. For video editing, Tune-A-Video (Wu et al., 2023)

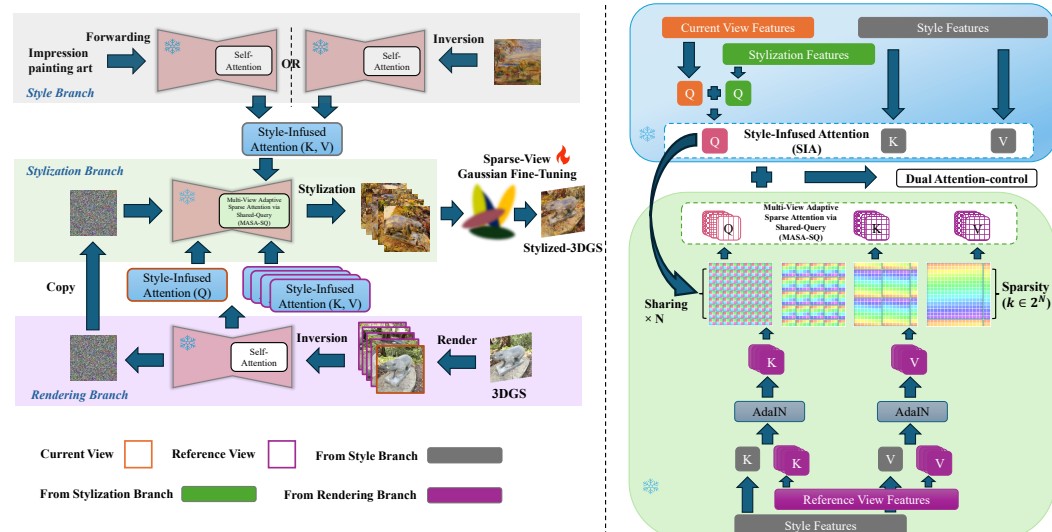

Figure 1: **Overview of the Diff-StyGS framework for multimodal 3D style transfer.** Our framework integrates three synergistic branches for consistent 3D stylization: **(Left) Style Branch** extracts style features (**K/V**) from input sources using a pre-trained DM; **Rendering Branch** provides content structure (**Q**) via 3DGS rendering; and **Stylization Branch** integrates both through a dual-attention mechanism. **(Right)** Specifically, **SIA** fuses content **Q** with style-derived **K/V** for style injection, while **MASA-SQ** ensures 3D consistency by sharing **Q** across views and modulating reference **K/V** via AdaIN in a sparse attention scheme.

adapts spatio-temporal attention to achieve temporally consistent edits, and CoDeF (Ouyang et al., 2024) proposes content deformation fields to propagate edits across motion dynamics. Further, InstructVideo (Yuan et al., 2024) leverages human feedback to align edits with complex instructions for interactive editing. These approaches highlight the flexibility of 2D DMs in handling various tasks, though challenges remain in extending control to 3D scenes with computational efficiency.

### 2.2 3D Generation, Editing and Style Transfer

3D scene modeling is typically built on implicit (Deng et al., 2022; Yuan et al., 2022) or explicit (Kwak et al., 2025; Lee et al., 2024) representations. NeRF (Mildenhall et al., 2020) trains neural networks for direct 3D scene rendering from multi-view images with camera locations. 3DGS (Kerbl et al., 2023) later surpasses NeRF in reconstruction quality and efficiency, using a set of 3D Gaussian distributions. Based on images as style inputs, FPRF (Kim et al., 2024), StyleRF (Liu et al., 2023) and StyleGaussian (Liu et al., 2024c) leverage 3D-aware feature representations for 3D neural style transfer. Alternatively, text-guided 3D scene stylization and editing, including GaussCtrl (Wu et al., 2024), DGE (Chen et al., 2024b), TIP-Editor (Zhuang et al., 2024) and ViCA-NeRF (Dong et al., 2023), could provide more user-friendly controls via text-image DM, at the cost of expansive DM finetuning with heavy computations (Brooks et al., 2023; Ye et al., 2023; Zhang et al., 2023). Furthermore, the success of 2D DM has recently motivated various solutions in 3D DM-based synthesis (Kim et al., 2023; Poole et al., 2023; Huang et al., 2024; Shi et al., 2024), consisting of 3D native generation, 2D prior-based 3D generation, and hybrid 3D generation. These advancements offer a promising avenue for controllable 3D editing with coherent multi-view modifications.

## 3 Methodology

### 3.1 Preliminaries

**Diffusion Model.** Diffusion-based generative models (Ho et al., 2020; Song et al., 2021a;b; Rombach et al., 2022) learn data distributions via a forward noising process and a reverse denoising process. In Latent Diffusion Models (LDMs), the forward process corrupts a latent representation $\mathbf{z}_0$ at step $t$ as $q(\mathbf{z}_t|\mathbf{z}_{t-1}) = \mathcal{N}(\mathbf{z}_t; \sqrt{1 - \beta_t}\mathbf{z}_{t-1}, \beta_t\mathbf{I})$, with noise schedule $\{\beta_t\}_{t=1}^T$. The reverse process trains a

denoiser $\boldsymbol{\epsilon}_\theta$ to predict noise in latent space, minimizing $\mathcal{L}_{\mathrm{DM}} = \mathbb{E}_{\mathbf{x}_0, c, t, \boldsymbol{\epsilon}} \left[ \|\boldsymbol{\epsilon}_\theta(\mathbf{x}_t, c, t) - \boldsymbol{\epsilon}\|^2 \right]$, where $c$ is the conditioning embedding. The U-Net architecture integrates a residual block for local feature transformation, a self-attention (SA) layer for spatial dependency modeling, and a cross-attention (CA) layer for condition injection. The attention is formulated by

$$\mathrm{Attn}(\mathbf{Q}, \mathbf{K}, \mathbf{V}) = \mathrm{Softmax}\left(\frac{\mathbf{Q}\mathbf{K}^\top}{\sqrt{d}}\right) \cdot \mathbf{V}, \tag{1}$$

where $\mathbf{Q}$ denotes *query* vectors projected from spatial features and $d$ is the embedding dimension of $\mathbf{Q}$. For SA modules, $\mathbf{K}$ (*key*) and $\mathbf{V}$ (*value*) derive from the same spatial features with $\mathbf{Q}$ (*query*) by projection matrices, while in CA layers, $\mathbf{K}$ and $\mathbf{V}$ are instead projected from conditional embeddings $c$. This work focuses on pre-trained text-to-image SD with text prompt conditioning $c$.

**3D Gaussian Splatting.** 3D Gaussian Splatting (3DGS) (Kerbl et al., 2023) explicitly and efficiently represents 3D scenes using anisotropic Gaussians, each parameterized by a mean $\mu \in \mathbb{R}^3$ and covariance $\mathbf{\Sigma} = \mathbf{R}\mathbf{S}\mathbf{S}^\top\mathbf{R}^\top$, where $\mathbf{R}$ and $\mathbf{S}$ are rotation and scaling matrices. The Gaussian function is $g_i(\mathbf{x}) = \exp\left(-\frac{1}{2}(\mathbf{x} - \mu_i)^\top \mathbf{\Sigma}_i^{-1}(\mathbf{x} - \mu_i)\right)$. For 2D rendering, Gaussians are projected via $\mathbf{\Sigma}_i' = \mathbf{J}_i \mathbf{W}_i \mathbf{\Sigma}_i \mathbf{W}_i^\top \mathbf{J}_i^\top$, with $\mathbf{W}_i$ as the viewing transformation and $\mathbf{J}_i$ as the projective Jacobian. Pixel color is computed as $C = \sum_{i=1}^N \mathbf{c}_i T_i \alpha_i g_i(\mathbf{x}_i)$, where $T_i = \prod_{j=1}^{i-1}(1 - \alpha_j g_j(\mathbf{x}_j))$, with $\mathbf{c}_i$ and $\alpha_i$ as the color and opacity of the $i$-th Gaussian.

### 3.2 FRAMEWORK OVERVIEW

Given a 3DGS represented by originally captured images and camera parameters, our objective is to achieve efficient and consistent style transfer from an arbitrary source (a 2D image or text prompt) onto the 3D scene with high fidelity. To this end, we propose Diff-StyGS, a framework that decouples the problem into two stages: ① zero-shot, multi-view consistent 2D image stylization via attention-based control in the frozen SD, and ② 3DGS adaptation based on the stylized images.

As illustrated in Fig. 1, the first stage integrates three coordinated branches to generate stylized views without training: (i) **Style Branch** extracts style representations (*key* $\mathbf{K}$ and *value* $\mathbf{V}$) from image inputs via DDIM inversion or text inputs via conditional encoding, preparing for content-aware stylization with our **Style-Infused Attention (SIA)**. (ii) **Rendering Branch** produces multi-view content renders from the original 3DGS, providing structural *query* $\mathbf{Q}$. (iii) **Stylization Branch** integrates these through dual attention mechanisms—SIA injects style features into self-attention layers, while **Multi-View Adaptive Sparse Attention via Shared-Query (MASA-SQ)** ensures cross-view geometric consistency by leveraging shared *query* across views & fused reference *key* and *value* via AdaIN (Huang et al., 2017) with dynamic attention sparsity. The second stage optimizes the 3DGS parameters under frequency-aware and view-aware supervision: **Wavelet Frequency Alignment Loss (WFAL)** enforces multi-frequency stylized alignment, and **Sparse-View Selection (SVS)** enhances efficiency by selecting geometrically diverse views in the camera-pose space.

### 3.3 STYLE-INFUSED ATTENTION

It has been revealed that CA layers are semantic bridges between input conditions and visual features, where spatial layouts and styles are altered by *key* and *value* (Hertz et al., 2023; Liu et al., 2024a). And SA inherently preserves the geometric and shape details of the source image (Tumanyan et al., 2023; Jiang et al., 2024). Building upon this insight, we unify the paradigm into SA layers by reinterpreting style latents from *key* and *value* as controllable conditions and maintaining content information from *query* features. Formally, given a 3DGS rendered input $\mathbf{I}_r$ to be edited and the multi-modal style source, we define $\mathbf{Q}_r^t$ as queries from $\mathbf{I}_r$ at diffusion step $t$, and $\mathbf{K}_s^t, \mathbf{V}_s^t$ (*key* and *value*) are derived from either ① *Style Image* by DDIM-inverted features, or ② *Text Prompt* via DDIM sampling. SA is reformulated as conditional attention named SIA (Style-Infused Attention):

$$\mathrm{SIA} = \mathrm{Attn}\left(\bar{\mathbf{Q}}_{sr}^t, \mathbf{K}_s^t, \mathbf{V}_s^t\right), \text{ where } \bar{\mathbf{Q}}_{sr}^t = \beta \cdot \mathbf{Q}_r^t + \eta \cdot \mathbf{Q}_{sr}^t. \tag{2}$$

$\beta$ and $\eta$ balance content preservation between original rendering query $\mathbf{Q}_r^t$ and stylized query $\mathbf{Q}_{sr}^t$ from the stylized generation. In standard SA, *query* and *key* features are from an identical sample. However, when replacing *key* with style-conditioned features, the inherent mismatch between content $\mathbf{Q}_r^t$ and style $\mathbf{K}_s^t$ would generate spatially smoothed attention maps (Chung et al., 2024). Thus, $\beta > 1$ and $\eta < 1$ scale SA for visually pleasant alignment.

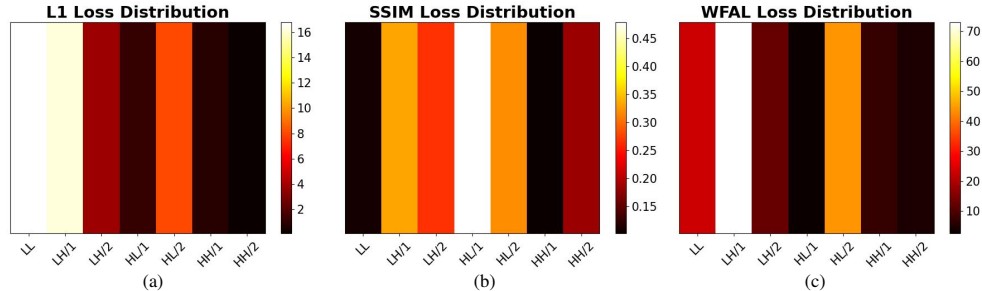

(a)  (b)  (c)

Figure 2: **Heatmap comparisons between $\mathcal{L}_1$, SSIM and WFAL** show the sensitivity of each loss in different frequency levels. (a) $\mathcal{L}_1$ loss strongly relies on global pixel value matching, thus having a higher sensitivity to the low-frequency sub-band (LL) and insensitive to high-frequency details (LH and HL). (b) SSIM focuses on structural similarity and improves perception of low to mid-frequency structures, but presents insufficient spectral alignment of high-frequency sub-bands. (c) In contrast, WFAL has clear differentiation in all frequency sub-bands, aligns the matching of mid to high-frequency spectra, and thus reduces sub-band errors ignored by $\mathcal{L}_1$ and SSIM.

### 3.4 MULTI-VIEW ADAPTIVE SPARSE ATTENTION VIA SHARED-QUERY

While prior works (Khachatryan et al., 2023; Li et al., 2024; Wu et al., 2023) leverage cross-view attention for temporal consistency, 3DGS stylization introduces unique challenges: ① view-dependent effects in 3D scenes amplify appearance discrepancies, and ② redundant attention across similar viewpoints leads to high computation on overlapping regions. To address these issues, we propose Multi-View Adaptive Sparse Attention via Shared-Query (MASA-SQ) to ensure multi-view consistency across reference views $\{\mathbf{I}_{\text{ref}}^i\}_{i=1}^{N_{ref}}$. Reference view features $\{\mathbf{K}_{\text{ref}}^i\}_{i=1}^{N_{ref}}$ and $\{\mathbf{V}_{\text{ref}}^i\}_{i=1}^{N_{ref}}$ are first aligned with style features via AdaIN (Huang et al., 2017) for preserving reference geometry and injecting style: $\mathbf{K}_{\text{fuse}}^i = \sigma(\mathbf{K}_s) \cdot \frac{\mathbf{K}_{\text{ref}}^i - \mu(\mathbf{K}_{\text{ref}}^i)}{\sigma(\mathbf{K}_{\text{ref}}^i)} + \mu(\mathbf{K}_s), \mathbf{V}_{\text{fuse}}^i = \sigma(\mathbf{V}_s) \cdot \frac{\mathbf{V}_{\text{ref}}^i - \mu(\mathbf{V}_{\text{ref}}^i)}{\sigma(\mathbf{V}_{\text{ref}}^i)} + \mu(\mathbf{V}_s)$, where $\mu(\cdot)$ and $\sigma(\cdot)$ denote channel-wise mean and std.

Irrelevant noise in multiple views affects 3D scene consistency, where adjacent viewpoints could induce more geometric conflicts (Liu et al., 2024d). This phenomenon motivates our approach to dynamically adjust the confidence weights assigned to different reference views based on their similarity to the current view to be edited. The normalized SSIM similarities between $\mathbf{I}_r$ and $\mathbf{I}_{\text{ref}}^i$ serve as view-specific attention weights $\{\omega_i\}_{i=1}^{N_{ref}} = \frac{\exp(\text{SSIM}(\mathbf{I}_r, \mathbf{I}_{\text{ref}}^i))}{\sum_{j=1}^{N_{ref}} \exp(\text{SSIM}(\mathbf{I}_r, \mathbf{I}_{\text{ref}}^j))}$. To further mitigate such non-semantic noisy information under fewer computing overheads, we introduce Grid-based Attention (Tu et al., 2022) with adaptive sparsity rates $\psi \in \{2^n\}_{n=1}^{N_{ref}}$, where SA is calculated merely between the same color patch divided by grid-size $2^n$ in Fig. 1. This implements a coarse-to-fine attention hierarchy, in which high-similarity views ($\omega_i \uparrow$) utilize sparser attention (grid-size $\uparrow$), while low-similarity ones ($\omega_i \downarrow$) employ denser patterns (grid-size $\downarrow$) to preserve details. MASA-SQ is implemented as follows:

$$\text{MASA-SQ} = \sum_{i=1}^{N_{ref}} \omega_i \cdot \text{GridAttn}_{\psi_i}(\mathbf{Q}_{\text{shared}}, \mathbf{K}_{\text{fuse}}^i, \mathbf{V}_{\text{fuse}}^i), \qquad (3)$$

where $\mathbf{Q}_{\text{shared}}$ is derived from Eq. (2). Finally, multi-view consistent stylized attention output, as the dual-attention control, is blended by SIA and MASA-SQ together:

$$\mathbf{\Phi}_{\text{final}} = \underbrace{\rho \cdot \text{SIA}}_{\text{Stylization}} + \underbrace{(1 - \rho) \cdot \text{MASA-SQ}}_{\text{View-coherence}}, \qquad (4)$$

where $\rho \in [0, 1]$ balances style preservation and multi-view consistency. The entire attention mechanism is presented in Algorithm 1 from Appendix B.

### 3.5 WAVELET FREQUENCY ALIGNMENT LOSS

Existing loss functions (*e.g.*, $\mathcal{L}_1$ and SSIM) in 3DGS training primarily emphasize global geometry and pixel-level matching while neglecting spectral alignment of high-frequency components, as illustrated in Fig. 2. This oversight could induce spectral distortions in reconstructed images, particularly for style transfer tasks. To achieve multi-scale spectral alignment

between low-frequency structures and high-frequency details, we propose Wavelet Frequency Alignment Loss (WFAL) as a complementary stylization loss. Given rendered and target features $\mathbf{F}_{\text{render}}, \mathbf{F}_{\text{tar}} \in \mathbb{R}^{C \times H \times W}$, WFAL first decomposes them via $L$-level Discrete Wavelet Transform (DWT): $\mathcal{W}(\mathbf{F}) = \left\{ \mathbf{A}_L, \{\mathbf{D}_l^{(k)}\}_{l=1}^L \right\}, k \in \{\text{LH}, \text{HL}, \text{HH}\}$, where $\mathbf{A}_L \in \mathbb{R}^{C \times H/2^L \times W/2^L}$ denotes the low-frequency approximation, and $\mathbf{D}_l^{(k)}$ represents high-frequency details at level $l$ along horizontal (LH), vertical (HL), and diagonal (HH) orientations. For each subband $\mathbf{S}_i \in \{\mathbf{A}_L, \mathbf{D}_l^{(k)}\}$, we then compute the Sliced Wasserstein Distance (SWD) between rendered and target distributions: $\text{SWD}_i = \mathbb{E}_{\theta \sim \mathcal{U}(\mathbb{S}^{d-1})} \left[ \mathcal{W}_1 \big( \texttt{sort}(\langle \mathbf{S}_i^{\text{render}}, \theta \rangle), \texttt{sort}(\langle \mathbf{S}_i^{\text{tar}}, \theta \rangle) \big) \right]$, where $\theta$ means random projection directions on the unit sphere $\mathbb{S}^{d-1}$, $\mathcal{W}_1$ is the 1-Wasserstein distance, and $\texttt{sort}$ ensures permutation-invariance. The final WFAL integrates multi-scale spectral alignment across all frequency sub-bands:

$$\mathcal{L}_{\text{WFAL}} = \underbrace{\text{SWD}_{\mathbf{A}_L}}_{\text{Low-Freq}} + \underbrace{\sum_{l=1}^L \sum_k \text{SWD}_{\mathbf{D}_l^{(k)}}}_{\text{High-Freq}}. \tag{5}$$

### 3.6 Sparse-View Selection

To reduce the computational cost and eliminate viewpoint redundancy, we propose a Maximin Distance based SVS strategy, which selects a sparse subset of images that are jointly optimized for spatial and rotational diversity in Fig. 3. Given any two cameras $c_i, c_j \in \mathcal{C}$, we define a composite distance metric as:

$$d(c_i, c_j) = \underbrace{\|\mathbf{p}_i - \mathbf{p}_j\|_2}_{\text{Positional Distance}} + \underbrace{\|\text{vec}(\mathbf{R}_i) - \text{vec}(\mathbf{R}_j)\|_2}_{\text{Rotational Distance}}, \tag{6}$$

where $\mathbf{p}_i \in \mathbb{R}^{3 \times 1}$ and $\mathbf{R}_i \in \mathbb{R}^{3 \times 3}$ denote the position vector and rotation matrix of camera $c_i$, respectively. The operator $\text{vec}(\cdot)$ flattens a matrix into a vector, ensuring consistent rotation dimensionality. To maximize viewpoint diversity, we iteratively select cameras via: $\mathcal{S}_{k+1} = \arg\max_{c \in \mathcal{C} \setminus \mathcal{S}_k} \left( \min_{c' \in \mathcal{S}_k} d(c, c') \right)$, where $\mathcal{S}_k$ represents the selected subset at iteration $k$. This ensures that each new camera $c$ is positioned to maximize the minimum distance to all previously selected cameras, thereby uniformly covering both spatial and angular domains. SVS effectively eliminates viewpoint redundancy with significantly reduced computation overhead.

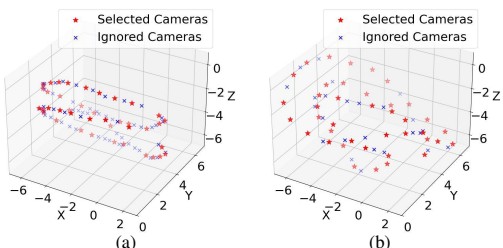

Figure 3: **Camera views in 3D space are sparsified by MDD.** We select representative views (*e.g.*, 40 cameras) from (a) 360° scenes and (b) forward-facing scenes.

### 3.7 Advantages

Our method effectively addresses the challenges of 3D style transfer with enhanced *multi-modal controllability*, *generative fidelity*, *multi-view consistency*, and *computational efficiency*. Specifically, SIA injects style features (such as content, layout, and style information from multi-modal inputs) into DM's attention, leading to enhanced controllability and fidelity. MASA-SQ shares the content *query* across views and adaptively fuses the reference *key/value*, improving generative view-consistency. WFAL further improves fidelity by enforcing wavelet distribution alignment. Meanwhile, multiple efficiency techniques are adopted, such as sparse attention in MASA-SQ, SVS, and frozen SD in SIA and MASA-SQ without fine-tuning.

## 4 Experiments

### 4.1 Experimental Setup

**Baselines and Implementation Details.** We compare our Diff-StyGS with state-of-the-art (SOTA) 3DGS stylization approaches, including text-guided diffusion frameworks DGE (Chen et al., 2024b)

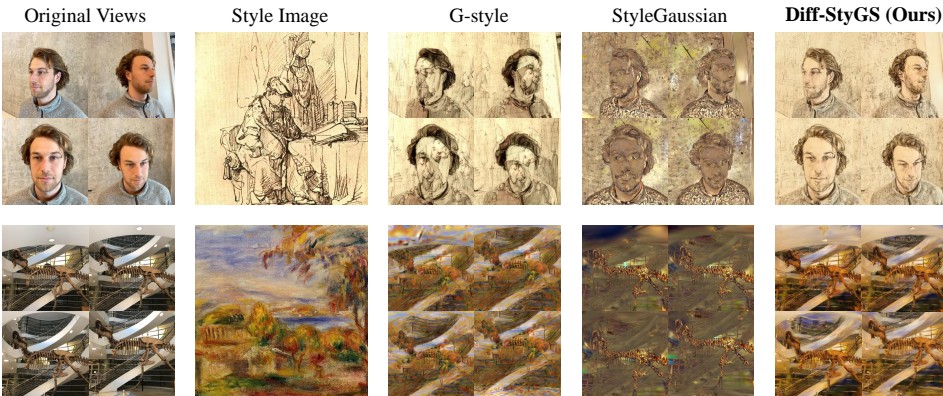

Figure 4: **Qualitative results on face-forwarding scenes using style images as inputs.** Our method generates higher-quality images with well-preserved content and visually appealing styles than other SOTA baselines. Unfortunately, both G-style (Kovács et al., 2024) and StyleGaussian (Liu et al., 2024c) fail to achieve satisfactory adaptions due to artifacts and immoderate styles.

using InstructPix2Pix (Brooks et al., 2023) and GaussCtrl (Wu et al., 2024) with ControlNet (Zhang et al., 2023), and image-driven neural stylizers G-style (Kovács et al., 2024) and StyleGaussian (Liu et al., 2024c). Although existing methods require either fine-tuned SD variants or style-aware neural matching, our framework introduces plug-and-play attention control over the original SD backbone. The 3DGS pipeline builds on NeRFStudio's splatfacto and vanilla SD-v2.1-base as our 2D stylization model with 50 timesteps. We set $\beta$ to 1.125, $\eta$ to 0.375 in Eq. (2) and $\rho$ to 0.65 in Eq. (4). We randomly sample 4 views as reference views and adopt MDD-based sparse-view selection for 40 frames. Eq. (5) is attached to traditional 3DGS training losses with a penalty strength of 0.01.

**Dataset and Evaluation Metrics.** Our experiments are constructed through various datasets, including two forward-facing scenes (Haque et al., 2023; Liu et al., 2025) and three 360° scenes (Barron et al., 2022; Haque et al., 2023). For image-guided stylization, we use the WikiArt dataset (Chung et al., 2024; Liu et al., 2024c) as the stylized source, while the text-driven paradigm utilizes concise semantic descriptions (*e.g.*, "Classical oil painting style"). All images from 3D scenes and WikiArt are center-resized to 512×512 for the resolution matching in SD. We establish a comprehensive evaluation protocol: For image-guided stylization, LPIPS (Zhang et al., 2018) quantifies content fidelity between 3DGS renderings and inputs, while FID (Heusel et al., 2017) measures style distribution divergence. ArtFID (Chung et al., 2024) combines these as a joint metric. In text-driven scenarios, CLIP Text-Image Direction Similarity (CLIP-TIDS) (Gal et al., 2022) evaluates a text-image alignment in CLIP space, complemented by CLIP Directional Consistency (CLIP-DC) (Haque et al., 2023) for cross-view embedding coherence. Multi-view consistency (MV-C) is measured via optical flow-warped RMSE (Liu et al., 2024c) across every two adjacent frames.

## 4.2 QUALITATIVE EVALUATION

Our Diff-StyGS demonstrates significant advantages in generating high-fidelity stylized 3D scenes across diverse scenarios, as shown in Fig. 4. The baselines, G-style (Kovács et al., 2024) and StyleGaussian (Liu et al., 2024c), introduce noisy artifacts and over-exaggerated content-style signals within the entire space, leading to underfitting and overfitting in 3DGS stylization. In contrast, our SIA module ensures style transfer fidelity by decoupling *key* & *value* projections from style features while retaining content structure via adaptive *query* fusion. As a result, our method achieves content-style disentanglement that preserves geometry while transferring artistic patterns.

For stylization guided by text prompts in 360° scenes, Diff-StyGS generates cross-view coherent stylization across complex backgrounds in Fig. 5. In detail, the "garden" in watercolor painting style by Diff-StyGS maintains color consistency in shadowed regions, whereas DGE (Chen et al., 2024b) generates artifacts and GaussCtrl (Wu et al., 2024) exhibits abrupt style shifts. This is attributed to our MASA-SQ, which enforces style consistency through shared queries and adaptive sparse grid attention. Furthermore, the proposed WFAL mitigates spectral distortions in high-frequency details. For instance, our Rococo-style "stone horse" retains sharp edge textures, while two baselines suffer

| Original Views | DGE | GaussCtrl | **Diff-StyGS (Ours)** |
|---|---|---|---|

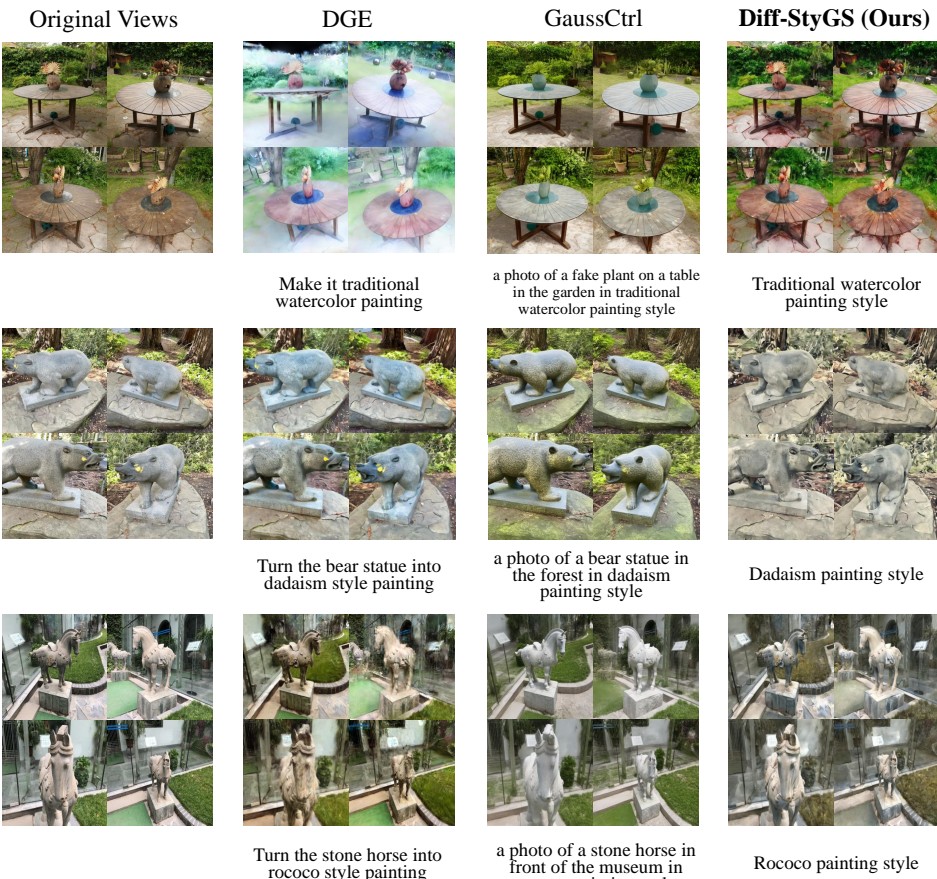

| Make it traditional watercolor painting | a photo of a fake plant on a table in the garden in traditional watercolor painting style | Traditional watercolor painting style |
|---|---|---|
| Turn the bear statue into dadaism style painting | a photo of a bear statue in the forest in dadaism painting style | Dadaism painting style |
| Turn the stone horse into rococo style painting | a photo of a stone horse in front of the museum in rococo painting style | Rococo painting style |

Figure 5: **Qualitative results on 360° scenes guided by text prompts.** The stylized 3D scenes rendered by Diff-StyGS yield more coherent views with pleasantness, compared with SOTA frameworks. Although DGE (Chen et al., 2024b) and GaussCtrl (Wu et al., 2024) are equipped with cross-view modules, they could lead to undesirable styles and coloring. Zoom in for better visual effects.

from blurring. These results validate that our unified framework—combining zero-shot attention mechanisms with frequency-aligned optimization under MDD-based SVS—achieves competing performance in multimodal 3D stylization tasks. To judge stylization more subjectively, we invite 15 raters for a user study, as detailed in Appendix C. It shows that Diff-StyGS achieves better visual fidelity and perceptual quality aligned with human observers.

| Method | Style Modality | 360° | | | | Forward-facing | | | |
|---|---|---|---|---|---|---|---|---|---|
| | | LPIPS↓ | FID↓ | ArtFID↓ | MV-C ×10³↓ | LPIPS↓ | CLIP-TIDS↑ | CLIP-DC↑ | MV-C ×10³↓ |
| G-style | Image | 0.69 | 19.48 | 34.61 | 48.93 | 0.65 | - | - | 48.14 |
| StyleGaussian | | 0.77 | 26.45 | 48.73 | 41.32 | 0.74 | - | - | 41.39 |
| Diff-StyGS (Ours) | | **0.52** | **18.86** | **30.19** | **40.87** | **0.50** | - | - | **40.66** |
| DGE | Text | 0.60 | - | - | 40.96 | 0.58 | 0.06 | 0.84 | 40.74 |
| GaussCtrl | | 0.58 | - | - | 39.27 | 0.57 | 0.06 | 0.85 | 38.63 |
| Diff-StyGS (Ours) | | **0.55** | - | - | **38.45** | **0.54** | **0.08** | **0.89** | **38.37** |

Table 1: **Performance comparison under different guidance modalities. Bold** numbers indicate the best results and arrows denote direction (↓=lower is better, ↑=higher is better). Diff-StyGS outperforms other baselines, in terms of cross-view consistency, stylization fidelity, and content perseveration. For each 3D scene, we run 3 experiments for stylization.

## 4.3 QUANTITATIVE EVALUATION

Although 3D style transfer is more subjective, we conduct quantitative evaluations to demonstrate the stylized effects from different perspectives, such as content-style fidelity, embedding coherence and

| Training Objectives | | | LPIPS↓ | |
|---|---|---|---|---|
| $\mathcal{L}_1$ | SSIM | WFAL | 360° | Forward-facing |
| ✔ | ✔ | ✗ | 0.56 | 0.53 |
| ✗ | ✗ | 0.01 | 0.65 | 0.62 |
| ✔ | ✔ | 0.01 | **0.53** | **0.51** |
| ✔ | ✔ | 0.1 | 0.69 | 0.67 |
| ✔ | ✔ | 0.5 | 0.74 | 0.70 |

(a)

(b)

Table 2: (a) Performance on stylization objectives in 3DGS. (b) Improved multi-view consistency could take the minimal computing cost when sparse attention is utilized in multiple frames.

multi-view consistency in Tab. 1. Diff-StyGS can support both style image inputs and text-prompt guidance, while the baselines can only support a single modal input. Furthermore, Diff-StyGS achieves the best performance on various evaluation metrics, compared with SOTA baselines across multiple 3D scenes with different complexities.

## 4.4 ABLATION STUDY

We explore the effectiveness of WFAL across various training settings, specifically excluding traditional $L_1$ and SSIM losses under 360° and forward-facing scenes. We also evaluate the impact of adjusting WFAL loss weights. As depicted in Tab. 2a, the incorporation of WFAL enhances traditional 3DGS loss functions, with the best performance when WFAL is weighted at 0.01. This highlights WFAL's role as a crucial supplement for improving stylization fidelity. Furthermore, Tab. 2b illustrates the efficiency gains achieved by employing adaptive sparse attention patterns within MASA-SQ. The computations are significantly reduced compared with multi-view dense attention (from $5\times$ FLOPs decreased to $1.08\times$), while maintaining competitive multi-view consistency with even better LPIPS. This demonstrates Diff-StyGS's capability to achieve superior generative performance efficiently, striking an effective balance between fidelity and computing resources. In Fig. 6, we visualize the stylized results under our dual-attention control by increasing attention layers in SD-v2.1-base. Since SD upsampling transformer blocks can capture style (Wang et al., 2024a;b), we begin at $7^{th}$ attention layer and add more layers until the last one ($11^{th}$) for the favorable style.

| Original | Style Image | Layer_7 | Layer_[7-8] | Layer_[7-9] | Layer_[7-10] | Layer_[7-11] |
|---|---|---|---|---|---|---|

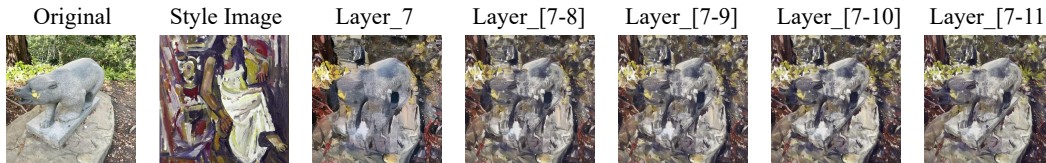

Figure 6: **Stylization results across layers in SD.** After adding more suitable layer-wise features into stylization dual-attention after $7^{th}$ layer, we can achieve more style-aligned images.

We provide additional ablation studies detailed in Appendix D to comprehensively validate key design choices in our framework, including (i) the number of selected views (which is set to 40) in SVS for better efficiency-quality trade-off, (ii) $\rho = 0.65$ in dual-attention control to achieve both stylistic fidelity and multi-view consistency, and (iii) style-content parameters ($\beta = 1.125$ and $\eta = 0.375$) to balance vivid stylization and structural integrity.

## 5 CONCLUSION

In this paper, we propose Diff-StyGS, a novel framework for high-quality multimodal 3DGS stylization through three innovations: ① a zero-shot multi-view attention mechanism that disentangles content-style interactions while ensuring cross-view consistency; ② WFAL for spectral-aligned style transfer via multi-scale frequency alignment; ③ MDD-based sparse-view selection for efficient optimization. Extensive experiments validate our framework's superiority in both image- and text-driven scenarios, and these advancements establish a new paradigm for 3D scene stylization. Future research could explore integrating 3D-aware generative models to enable joint editing of geometry and appearance, thereby overcoming current constraints and expanding the scope of 3D scene manipulation.

## 6 REPRODUCIBILITY STATEMENT

We have made comprehensive efforts to ensure the reproducibility of our work. Detailed descriptions of our experimental setup, including hyperparameter settings and environment packages, are provided in Section 4. The proposed dual-attention control is formally outlined in Algorithm 1. To further support replication, all source code will be made publicly available upon acceptance of the paper.

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

APPENDIX

In this section, we provide the following:

◇ The use of Large Language Models (LLMs).

◇ Pseudo code for our proposed dual-attention control.

◇ User Study for more subjective evaluation.

◇ More experimental ablation results.

◇ Additional visualizations and comparisons between Diff-StyGS and other baselines.

## A  USAGE OF LLMS

The authors use LLMs solely for the final proofreading stage, to improve the writing's clarity and fluency by identifying and correcting grammatical errors and typos. The LLMs are not applied to formulate ideas and develop the methods; all intellectual contributions remain entirely the work of the human authors.

## B  PSEUDO CODE

---

**Algorithm 1** Dual-Attention Control for Zero-shot, Multi-view Consistent Stylization

---

**Require:**

Target view render $\mathbf{I}_r$, diffusion step $t$, Style source (image/text), reference views $\{\mathbf{I}_{\text{ref}}^i\}_{i=1}^{N_{\text{ref}}}$

Hyperparameters: $\beta, \eta, \rho$

**Step 1: Feature Extraction**

1: $\mathbf{K}_s^t, \mathbf{V}_s^t \leftarrow \text{ExtractStyleFeatures}(\text{style}, t)$ ▷ Style Branch

2: $\mathbf{Q}_r^t \leftarrow \text{Encode}(\mathbf{I}_r)$ ▷ Rendering Branch

3: $\mathbf{Q}_{sr}^t \leftarrow \text{StylizedQuery}(\mathbf{I}_r, \text{style}, t)$

**Step 2: Style-Infused Attention (SIA)**

4: $\bar{\mathbf{Q}}_{sr}^t \leftarrow \beta \cdot \mathbf{Q}_r^t + \eta \cdot \mathbf{Q}_{sr}^t$ ▷ Content-preserving blend

5: $\text{SIA} \leftarrow \text{Attn}\big(\bar{\mathbf{Q}}_{sr}^t, \mathbf{K}_s^t, \mathbf{V}_s^t\big)$

**Step 3: Multi-View Adaptive Sparse Attention (MASA-SQ)**

6: $\mathbf{Q}_{\text{shared}} \leftarrow \bar{\mathbf{Q}}_{sr}^t$ ▷ Shared query across views

7: $\text{MASA-SQ} \leftarrow \mathbf{0}$

8: $\mathbf{s} \leftarrow [\,]$ ▷ Initialize SSIM scores array

9: **for** $i = 1$ to $N_{\text{ref}}$ **do**

10:   $\mathbf{s}[i] \leftarrow \text{SSIM}(\mathbf{I}_r, \mathbf{I}_{\text{ref}}^i)$ ▷ Compute similarity for each view

11: **end for**

12: $\{\omega_i\}_{i=1}^{N_{\text{ref}}} \leftarrow \text{Softmax}(\mathbf{s})$ ▷ Normalize to probability distribution

13: indices $\leftarrow \text{Argsort}(\{\omega_i\}_{i=1}^{N_{\text{ref}}}, \text{descending} = \text{True})$ ▷ Sort by weight descending

14: **for** rank $= 1$ to $N_{\text{ref}}$ **do**

15:   $i \leftarrow \text{indices}[\text{rank}]$ ▷ Get original index for this rank

16:   $\mathbf{K}_{\text{ref}}^i, \mathbf{V}_{\text{ref}}^i \leftarrow \text{Encode}(\mathbf{I}_{\text{ref}}^i)$

17:   $\mathbf{K}_{\text{fuse}}^i, \mathbf{V}_{\text{fuse}}^i \leftarrow \text{AdaIN}(\mathbf{K}_{\text{ref}}^i, \mathbf{V}_{\text{ref}}^i; \mathbf{K}_s^t, \mathbf{V}_s^t)$

18:   $\psi_i \leftarrow 2^{N_{\text{ref}}-\text{rank}+1}$ ▷ Adaptive sparsity: $\psi_i \in \{2^n\}_{n=1}^{N_{\text{ref}}}, \omega_i \uparrow \Rightarrow \psi_i \uparrow$

19:   $\text{MASA-SQ} \leftarrow \text{MASA-SQ} + \omega_i \cdot \text{GridAttn}_{\psi_i}(\mathbf{Q}_{\text{shared}}, \mathbf{K}_{\text{fuse}}^i, \mathbf{V}_{\text{fuse}}^i)$

20: **end for**

**Step 4: Dual-Attention Fusion**

21: $\boldsymbol{\Phi}_{\text{final}} \leftarrow \rho \cdot \text{SIA} + (1-\rho) \cdot \text{MASA-SQ}$ ▷ Stylization Branch

22: **return** $\boldsymbol{\Phi}_{\text{final}}$

---

## C  USER STUDY

To evaluate the perceptual quality and subjective appeal of our proposed Diff-StyGS, we conduct a user study against four SOTA baselines: StyleGaussian (Liu et al., 2024c), G-Style (Kovács et al., 2024), DGE (Chen et al., 2024b), and GaussCtrl (Wu et al., 2024). The rendering views for the study are derived from a set of five scenes, including both complex 360° environments and forward-facing scenes, each with a text prompt and an image as a style source. We invite 15 participants to join this study. In each trial, participants are presented with the stylized results from all five methods (Diff-StyGS and the four baselines) in a randomized order to mitigate ordering bias. They are then asked to select the single best result for each of the following three criteria: ① Content Preservation, ② Style Fidelity, and ③ Overall Quality & Consistency. This process yields a total of 150 preference feedback for each evaluation axis (10 stylized scenes × 15 participants). As summarized in Fig. 7, the overall preference rates reveal a clear and consistent superiority of our Diff-StyGS, providing strong empirical evidence that our approach generates artistically stylized renderings with a markedly higher degree of visual fidelity and perceptual quality as judged by human observers.

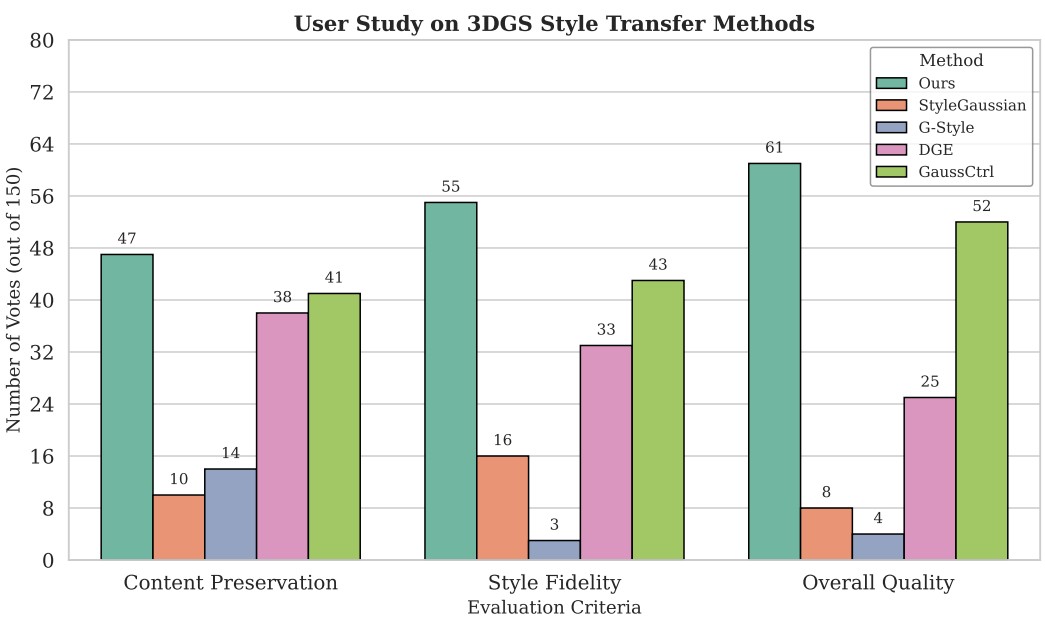

Figure 7: **User Study.** Diff-StyGS receives significantly higher user preference ratings for content preservation, style fidelity, and overall visual quality than competing baselines.

## D  EXTRA ABLATION STUDIES

**Impact of sparse views on stylization.** We investigate the influence of view selection on stylization fidelity and optimization efficiency. As quantified in Tab. 3a, Diff-StyGS achieves the optimal trade-off between rendering quality and stylization speed when leveraging 40 views selected via MDD.

**Balancing stylized fidelity and multi-view consistency.** To analyze the role of the coefficient $\rho$ in dual-attention control, we conduct experiments with $\rho \in [0.5, 0.65, 1.0]$. $\rho$ balances stylization fidelity (from SIA) against multi-view structural consistency (from MASA-SQ). Tab. 3b demonstrates that tuning $\rho$ offers direct control over stylistic strength and geometric coherence. Among the tested settings, $\rho = 0.65$ delivers the best trade-off, and is used as the default throughout our pipeline.

**Style-content preservation.** We conduct an ablation study to analyze the influence of the style strength parameter $\beta$ and the content preservation weight $\eta$. As illustrated in Fig. 8, these parameters jointly control the trade-off between style integration and content fidelity. Specifically, an excessively

| Views | LPIPS↓ | Speed-Up↑ |
|-------|--------|-----------|
| All   | 0.57   | 1×        |
| 20    | 0.74   | 6.7×      |
| 40    | **0.58** | 3.9×    |

(a)

| $\rho$ | LPIPS↓ | MV-Consist.$_{\times 10^3}$↓ |
|--------|--------|------------------------------|
| 1.0    | 0.57   | 40.08                        |
| 0.65   | **0.58** | 39.57                      |
| 0.5    | 0.64   | 39.49                        |

(b)

Table 3: (a) Performance on sparse-view stylization in 3DGS from 360° and forward-facing scenes. (b) The trade-off between stylization quality and multi-view consistency.

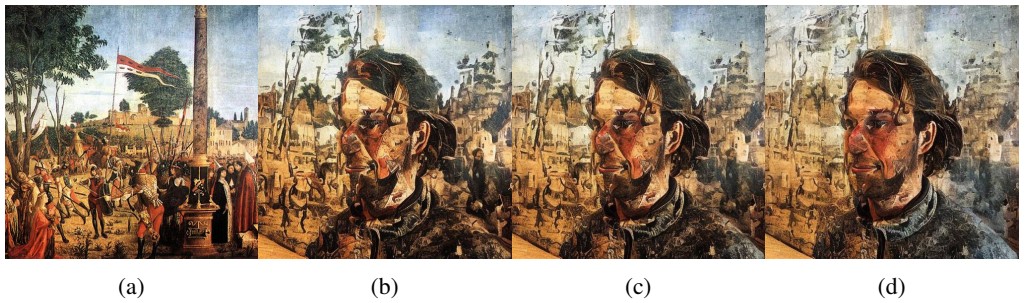

(a)       (b)       (c)       (d)

Figure 8: (a) Source style image. (b) $\beta = 0.375, \eta = 1.125$. (c) $\beta = 0.825, \eta = 0.675$. (d) $\beta = 1.125, \eta = 0.375$.

high $\eta$ and lower $\beta$ can lead to over-stylization, reducing the constraint of maintaining the original scene's structure and content in Fig. 8b and Fig. 8c. Based on empirical evaluation, we select $\beta = 1.125, \eta = 0.375$ as our default setting, as it strikes a compelling balance between vivid style expression and faithful content preservation.

# E ADDITIONAL VISUALIZATIONS

Our method shows notable improvements in high-quality 3D scene stylizations across various scenarios (Barron et al., 2022; Haque et al., 2023; Liu et al., 2025). As shown in Fig. 9, our Diff-StyGS maintains geometry and transfers rational styles for face-forwarding scenes. In contrast, G-style (Kovács et al., 2024) and StyleGaussian (Liu et al., 2024c) generate unpleasant artifacts and highly overblown contents and styles in the images. Regarding text-prompt guided 360° scene stylizations, Diff-StyGS follows the textual guidance well and keeps cross-view consistency when style transferring across different backgrounds in Fig. 10. However, DGE (Chen et al., 2024b) can introduce uncomfortable colors and noises to the entire image, while GaussCtrl (Wu et al., 2024) appears to disregard the text instructions and generate undesired results.

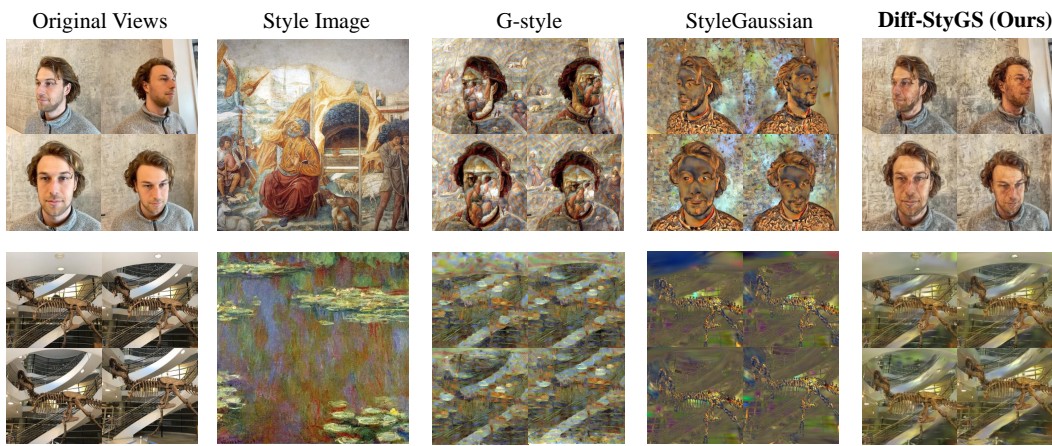

Figure 9: **Qualitative results on face-forwarding scenes using style images as inputs.** Our method preserves content and transfers pleasant styles than other SOTA baselines (Kovács et al., 2024; Liu et al., 2024c).

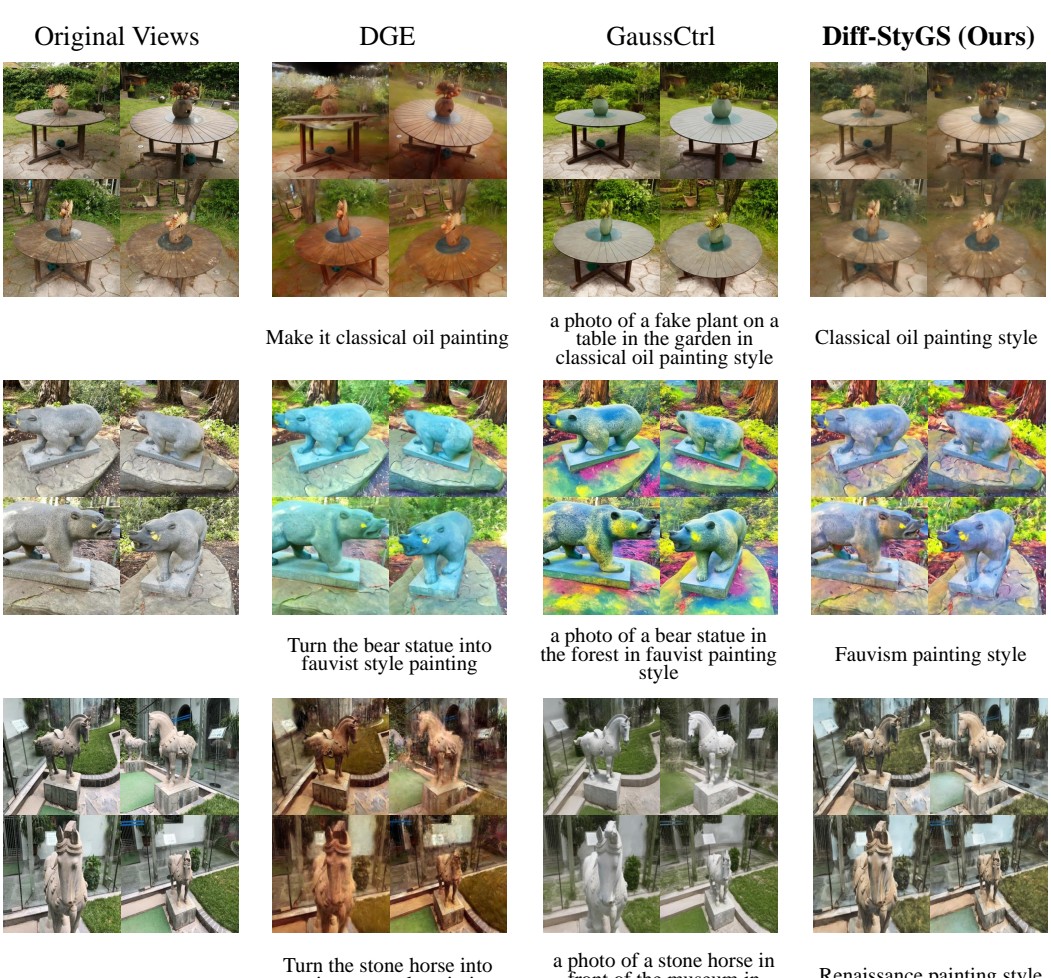

Figure 10: **Qualitative results on 360° scenes guided by text prompts.** The 3D scenes stylized by Diff-StyGS render more text-aligned views with multi-view coherence than previous SOTA methods (Wu et al., 2024; Chen et al., 2024b).

