# OpenReview forum: "Diff-StyGS: 3D Gaussian Splatting Stylization via Tuning-Free Multi-View Sparse Diffusion"
_ICLR.cc/2026/Conference — ICLR 2026 Conference Withdrawn Submission_

### Official Review · Reviewer_u8rm · 2025-10-28

**Soundness:** 2
**Presentation:** 3
**Contribution:** 2
**Rating:** 2
**Confidence:** 4

**Summary:**

This paper presents Diff-StyGS, a novel framework for performing 3D style transfer on 3D Gaussian Splatting (3DGS) representations. The core approach utilizes a tuning-free multi-view sparse diffusion mechanism to stylize the 3D scene while aiming to maintain cross-view consistency and computational efficiency. The method proposes several novel components, including Style-Infused Attention, a Wavelet Frequency Alignment Loss (WFAL), and a Multi-View Adaptive Sparse Attention via Shared-Query (MASA-SQ) to guide the stylization process across views. Experiments showcase the method’s ability to generate visually consistent stylized 3D scenes.

**Strengths:**

1. **Novel Integration**: The paper introduces an interesting and potentially impactful approach to 3DGS stylization by integrating a tuning-free diffusion mechanism with a multi-view strategy, which addresses the common issues of cross-view inconsistency and high computational cost associated with fine-tuning large diffusion models.

2. **Effective Stylization Across Modalities**: The qualitative results for both image-based and text-based style transfer appear effective, generating realistic and stable stylized renderings across different views.

3. **Cross-View Consistency**: The design inherently aims to enforce consistency by working with sparse, multi-view constraints, a critical challenge in 3D-aware style transfer.

**Weaknesses:**

1. **Limited Stylization Quality Improvement**: The overall improvement in stylization quality compared to current state-of-the-art methods is marginal. Specifically, in image-based style transfer, the stylization quality of baselines like G-Style appears better or at least comparable to the proposed Diff-StyGS. Furthermore, in text-based style transfer, the stylization quality shows no significant practical improvement, remaining similar to existing methods like DGE and GaussCtrl.

2. **Insufficient Qualitative Ablation Studies**: The evaluation lacks critical qualitative ablation studies for key proposed components such as the WFAL and the MASA-SQ. Given that style transfer is a highly subjective task, qualitative results are essential to validate the contribution of each module. Furthermore, the paper’s interpretation of LPIPS as a primary metric is questioned, as a higher LPIPS does not consistently guarantee subjectively "better" stylization results.

3. **Missing Computational Efficiency Experiments**: Despite the paper claiming computational efficiency as a key motivation and benefit, no experimental data is provided comparing the stylization time of Diff-StyGS against the baselines. This omission weakens a central claim of the work.

4. **Confusing Distance Metric in Equation 6**: The formulation of the distance computation in Eq. 6, which appears to sum a position vector and a rotation matrix, is confusing. These components operate on different scales and normalizations. The authors must clarify the exact distance being computed, whether the position vector is normalized, and provide a clear rationale for this simple additive combination.

5. **Lack of Comparison to Uniform Sampling**: The paper does not provide an experimental comparison between the proposed sparse-view selection strategy and a simple uniform sampling selection method. This makes it difficult to ascertain the actual benefit of the more complex selection logic.

6. **Ambiguous Figure 2 Comparison**: The comparison presented in Figure 2 is unclear. The visual representation suggests that SSIM might offer better coverage of frequency sub-bands, which may contradict the point the authors intended to make about their proposed method. This figure requires clarification or re-evaluation with another metric.

**Questions:**

The algorithm relies on several key hyperparameters ($\beta$, $\eta$, and $\rho$). Can the authors provide a comprehensive analysis (e.g., sensitivity study or sweep) demonstrating how these hyperparameters affect the final stylization quality? This analysis is crucial for understanding the effect and interplay of each component within the overall optimization process.

---

### Official Review · Reviewer_YmSw · 2025-10-30

**Soundness:** 2
**Presentation:** 2
**Contribution:** 2
**Rating:** 2
**Confidence:** 4

**Summary:**

This paper proposes Diff-StyGS, a two-stage 3D style transfer framework:
Stage 1: Achieves training-free, multi-view consistent 2D stylized image generation through Style-Infused Attention (SIA) and Multi-View Adaptive Sparse Attention via Shared-Query (MASA-SQ).
Stage 2: Optimizes the 3D Gaussian scene using Wavelet Frequency Alignment Loss (WFAL) and Sparse-View Selection (SVS) to enhance style fidelity and training efficiency.
The method supports both text and image style inputs, and outperforms existing state-of-the-art approaches across multiple datasets and scenarios.

**Strengths:**

1. Method Contribution. The authors adeptly adapt and combine existing techniques—such as attention control, style transfer, and wavelet losses—to address the specific challenges of 3D style transfer (e.g., multi-view consistency, spectral distortion). The MASA-SQ module, in particular, effectively tackles the cross-view consistency problem through its shared-query and adaptive sparsification mechanism.

2. Flexible Multi-Modal Control. The framework supports both reference images and text prompts as style inputs, offering users more flexible and intuitive control compared to existing methods that are often limited to a single modality.

3. Highly Efficient Training-Free Pipeline. The entire pipeline is training-free, leveraging a frozen pre-trained diffusion model and avoiding the need for computationally expensive fine-tuning. The incorporation of Sparse-View Selection (SVS) and adaptive sparse attention reduces computational and memory overhead, enabling highly efficient optimization.

**Weaknesses:**

1. Less of technical contributions: The primary contribution of this paper lies in its well-designed and efficient integrated framework, rather than in the raw novelty of its foundational components, like attention and diffusion models.

2. Dependence on Pre-trained Models: The method relies on a pre-trained Stable Diffusion model, whose generative capabilities and inherent biases may influence the final results.

3. Generalization to Complex Scenes: Although experiments cover diverse scenarios, their performance on extreme or unstructured scenes requires further validation.

4. Insufficient Comparison with Some Recent Works: The comparison with some of the latest 3D style transfer methods is not comprehensive enough, like GaussianEditor(CVPR2024), CTRL-D(CVPR2025), DreamCatalyst(ICLR2025), or those based on NeRF or GAN methods, should be compared.

5. Insufficient Control of Variables, Undermining Fair Comparison: In Figure 5 (Qualitative results on 360° scenes guided by text prompts), why does the paper employ different input text prompts for different compared methods? The use of different prompts for different methods undermines the fairness of the comparison, as the output is highly sensitive to the prompt wording.

**Questions:**

1. Is the design of SIA and MASA-SQ dependent on a specific diffusion model architecture (e.g., SD v2.1)? How generalizable is it to other diffusion models?

2. What is the advantage of using Sliced Wasserstein Distance (SWD) in WFAL compared to other distribution alignment methods, such as Maximum Mean Discrepancy (MMD)? Were any ablation studies conducted to justify this choice?

3. For the Sparse-View Selection (SVS), have the authors considered content-based or semantic-aware selection strategies, rather than relying solely on camera pose geometry?

4. Can the method support stylization of dynamic 3D scenes (e.g., using dynamic Gaussian splatting)? Is there a plan to extend this work to 4D content in the future?

5. Is it possible to provide comparison results for all methods using strictly identical text prompts? This would be essential to ensure that the observed performance differences are indeed attributable to the core algorithms rather than variations in the input conditions.

---

### Official Review · Reviewer_m5n1 · 2025-11-01

**Soundness:** 3
**Presentation:** 3
**Contribution:** 3
**Rating:** 6
**Confidence:** 3

**Summary:**

This paper proposes Diff-StyGS, a multimodal style transfer framework based on 3D Gaussian Splatting (3DGS). The method achieves style transfer through tuning-free Stable Diffusion models, primarily incorporating two innovative modules: (1) Style-Infused Attention (SIA); (2) Multi-View Adaptive Sparse Attention via Shared-Query (MASA-SQ). Additionally, the paper introduces Wavelet Frequency Alignment Loss (WFAL) and a Maximin Distance-based Sparse-View Selection (SVS) strategy. Experiments demonstrate that the method outperforms existing approaches in terms of multi-view consistency, stylization quality, and content fidelity.

**Strengths:**

1. Style-Infused Attention (SIA) effectively decouples content and style information.

2. Multi-View Adaptive Sparse Attention via Shared-Query (MASA-SQ) ensures cross-view consistency.

3. The paper proposes Wavelet Frequency Alignment Loss (WFAL) and Maximin Distance-based Sparse-View Selection (SVS) strategy.

**Weaknesses:**

1. Lack of theoretical analysis on the attention mechanism design choices. Why is the combination of SIA and MASA-SQ optimal?

2. Key parameters in WFAL, such as the number of projection directions for Sliced Wasserstein Distance and the wavelet decomposition level L, lack sensitivity analysis.

3. The choice of 40 views for sparse-view selection lacks sufficient theoretical or experimental justification.

4. The selection rationale for certain hyperparameters (β=1.125, η=0.375, ρ=0.65) is insufficiently supported.

**Questions:**

1. How do you ensure that 4 reference views can represent the entire 3D scene? Have you considered adaptively selecting the number of reference views?

2. How does MASA-SQ perform under extreme viewpoint changes (e.g., switching from top view to bottom view)?

3. How is the wavelet decomposition level L determined? Do different scenes require different L values?

4. Is the selection of 40 views optimal for all scenes? Do scenes with different complexity levels require different numbers of views?

5. Under what circumstances would this method fail? For example, for extremely abstract styles or styles requiring substantial geometric deformation?

---

### Official Review · Reviewer_8pX2 · 2025-11-04

**Soundness:** 2
**Presentation:** 2
**Contribution:** 2
**Rating:** 4
**Confidence:** 4

**Summary:**

The paper introduces Diff-StyGS, a tuning-free framework for 3D style transfer in 3D Gaussian Splatting (3DGS) using multimodal inputs (text or image). Diff-StyGS enables text- and image-based style transfer for 3DGS without requiring model fine-tuning or custom style-specific components. Experiments conducted on two forward-facing scenes and three 360-degree scenes demonstrate the effectiveness of the proposal.

**Strengths:**

S1. The storyline of this paper is clear and easy to follow.

S2. 3D Gaussian Splatting stylization is an interesting topic.

S3. The proposed multi-view stylized attention is reasonable.

**Weaknesses:**

W1. The paper only conducts ablation studies on loss functions in Section 4.4. There are no ablation studies on the proposed Style-infused attention, multi-view adaptive sparse attention via shared query. The effectiveness of each sub-component of the proposed framework is not clear.

W2. Leveraging query, key, and value in the attention mechanism of diffusion models for style transfer has been widely used in the literature. Sharing query for consistent image generation has also been explored in consistent text-to-image generation. Frequency-based loss functions are also widely used in various tasks. Therefore, I think the technological innovation of this paper is somewhat limited and did not reach the bar of ICLR.

W3. The authors didn’t provide video results of the stylized 3D Gaussian. It’s necessary to provide a video comparison of the results in the supplementary material.

**Questions:**

Please see the weakness.

---

### Note · Authors · 2025-11-14

I have read and agree with the venue's withdrawal policy on behalf of myself and my co-authors.